# Advancing Thoracic Surgical Oncology in the Era of Precision Medicine

**DOI:** 10.3390/cancers17010115

**Published:** 2025-01-02

**Authors:** Giacomo Argento, Erino Angelo Rendina, Giulio Maurizi

**Affiliations:** Department of Thoracic Surgery, University of Rome La Sapienza, Sant’Andrea Hospital, 00189 Rome, Italy; giacomo.argento@uniroma1.it (G.A.); erinoangelo.rendina@uniroma1.it (E.A.R.)

**Keywords:** lung cancer, precision medicine, thoracic surgery, genomics, proteomics, next-generation sequencing, multi-omic profiling

## Abstract

This paper highlights how advancements in precision medicine are transforming thoracic surgical oncology. By incorporating detailed molecular data, such as genetic and protein analyses, the study explores new ways to tailor cancer treatments for each patient. The authors focus on how these approaches enhance surgical planning and improve outcomes, especially for complex lung and airway cancers. Emerging technologies, such as next-generation sequencing and molecular imaging, provide deeper insights into tumor behavior, enabling personalized and effective treatment strategies. These findings aim to guide clinicians in using precision medicine to achieve better patient care and set the stage for future research to further improve the field.

## 1. Introduction

The management of thoracic malignancies, including non-small cell lung cancer (NSCLC), once relied predominantly on anatomical staging, histopathology, and performance status. Over the past decade, however, the landscape has evolved dramatically. Precision medicine allows clinicians to dissect tumor biology with unprecedented granularity, offering insights that inform both systemic therapies and surgical decisions. This paradigm shift acknowledges that thoracic tumors are highly heterogeneous entities, shaped by distinct genetic alterations, epigenetic modifications, complex signaling networks, and intricate immune microenvironments.

As this molecular complexity becomes clearer, tools such as next-generation sequencing (NGS), proteogenomic integration, and multi-omic profiling enable the identification of actionable targets and predictive biomarkers. In parallel, immunotherapy—guided by PD-L1 expression, TMB, and other emerging markers—has reshaped the perioperative treatment landscape. Surgical strategies now adapt to these insights: a tumor that responds dramatically to targeted neoadjuvant therapy might require less extensive resection, whereas a tumor with a more aggressive molecular profile may necessitate broader intervention.

Guidelines from the National Comprehensive Cancer Network (NCCN) [1], European Society for Medical Oncology (ESMO) [2], and other authoritative bodies increasingly reflect these changes. For example, routine testing for EGFR, ALK, ROS1, and PD-L1 in advanced NSCLC is now standard. As technology matures, it is plausible that multi-omic signatures, ctDNA analysis for minimal residual disease (MRD), and epigenetic markers will find their way into these guidelines, pushing the boundaries further. However, the extent to which these newer modalities are integrated into standard treatment plans varies among institutions, and questions remain regarding their cost-effectiveness, accessibility, and long-term benefits.

Yet, significant challenges remain. Integrating large volumes of molecular data into daily practice is non-trivial, requiring robust bioinformatics pipelines, clear interpretation standards, and continuous education for clinicians. Ethical questions—around privacy, consent, equitable access, and potential genetic discrimination—are now part of the thoracic oncology conversation. Finally, the field is dynamic: as new biomarkers are discovered and new drugs approved, we must anticipate a rapidly evolving standard of care. This review aims to provide a comprehensive overview of how precision medicine principles are being integrated into thoracic surgical oncology, highlighting opportunities, barriers, and future directions.

## 2. Precision Surgery and Oncogenomics in Thoracic Cancer

Oncogenomics focuses on cancer-related genetic alterations, forming the foundation of precision oncology. NGS, a high-throughput technology that allows rapid sequencing of entire genomes or targeted regions, enables clinicians to decipher the genetic landscape of thoracic tumors, thereby guiding the selection of targeted therapies and optimizing surgical strategies [3].

NGS can identify actionable mutations, such as EGFR, KRAS, and ALK mutations, that drive tumor growth, enabling more precise and effective treatment decisions [4]. For example, EGFR mutations may prompt the use of EGFR inhibitors like osimertinib [5], reducing tumor size, thereby allowing for more conservative surgical interventions.

On the other hand, KRAS mutations often indicate aggressive tumor behavior, necessitating more extensive resections to minimize recurrence risk [6,7].

The impact of this genetic understanding on preoperative planning is profound. The presence or absence of actionable mutations can determine the suitability of neoadjuvant therapies before surgery, ultimately optimizing outcomes. For example, in clinical practice, patients with EGFR-mutant NSCLC have shown substantial tumor shrinkage after neoadjuvant EGFR inhibition, facilitating less extensive surgical resections [8]. Similarly, the advent of KRAS G12C inhibitors has opened avenues for preoperative molecular control of tumors once considered challenging, potentially simplifying surgical procedures [9]. Furthermore, ALK rearrangements often respond well to ALK inhibitors, enabling tumor reduction and more conservative surgical approaches [10]. Additionally, MET exon 14 skipping mutations can be targeted using MET inhibitors like capmatinib, enhancing resectability [11]. ROS1 rearrangements, although less common, can be treated with inhibitors like crizotinib, allowing tumor downstaging [12]. Such genetic insights facilitate the application of precise neoadjuvant therapies, expanding treatment options and improving surgical outcomes for patients with thoracic cancers. Overall, identifying these actionable mutations allows for more targeted interventions, improving the precision and efficacy of both neoadjuvant and surgical treatment strategies.

These genomic insights align with and expand upon current guidelines from bodies like NCCN and ESMO [1,2], which already recommend routine testing for EGFR, ALK, and ROS1 in advanced NSCLC. As evidence accumulates for emerging biomarkers, we may see formal incorporation of these neoadjuvant strategies into standard treatment algorithms, guiding thoracic surgeons toward more personalized operative plans.

Beyond targeting mutations, NGS allows clinicians to evaluate tumor mutational burden (TMB) and microsatellite instability (MSI), factors that influence responses to immunotherapies. High-TMB or MSI-high tumors are often more likely to respond to immune checkpoint inhibitors, such as pembrolizumab, enhancing preoperative tumor shrinkage and increasing the probability of complete surgical resection [13].

Furthermore, the evaluation of programmed death-ligand 1 (PD-L1) expression has become a crucial component in guiding immunotherapy decisions. Immune checkpoint inhibitors targeting the PD-1/PD-L1 axis, such as nivolumab or atezolizumab, have demonstrated significant efficacy in patients with high PD-L1 expression, leading to prolonged disease-free intervals and improved survival outcomes [14]. In the neoadjuvant setting, combining immune checkpoint inhibitors with chemotherapy has shown improved pathological complete response rates, offering a powerful adjunct to surgery for high-risk patients. Notably, several clinical trials have contributed to shaping these strategies [Table 1]. For example, the CheckMate 816 trial demonstrated that neoadjuvant nivolumab combined with chemotherapy significantly improved the pathological complete response rate compared to chemotherapy alone in resectable NSCLC [15]. Additionally, the NADIM trial highlighted the benefits of combining nivolumab with chemotherapy in locally advanced NSCLC, showing significant improvements in overall survival and response rates [16]. Building upon these findings, the NADIM II trial further confirmed the efficacy of neoadjuvant chemoimmunotherapy, showing a substantial increase in pathological complete response rates and two-year overall survival, reinforcing the role of immunotherapy in the preoperative management of resectable NSCLC [17].

The ongoing NeoADAURA trial is investigating osimertinib in the neoadjuvant setting for patients with EGFR-mutated NSCLC, further expanding the scope of precision medicine in early-stage disease [18].

Together, these clinical trials illustrate the ongoing evolution towards a more individualized treatment paradigm in thoracic oncology, emphasizing the importance of molecular and immunologic profiling to optimize treatment selection and sequencing for improved patient outcomes.

## 3. Proteogenomics and Multi-Omic Integration for Comprehensive Care

Proteogenomics integrates proteomic data with genomic sequencing to provide a nuanced view of tumor biology [19]. By analyzing gene mutations and their downstream effects—such as protein overexpression, phosphorylation, and pathway activation—clinicians can more accurately predict a tumor’s behavior and response to therapies. Technologies like mass spectrometry allow the identification of aberrant proteins and post-translational modifications that serve as therapeutic targets or prognostic markers, offering insights beyond genomic data alone.

In NSCLC patients with EGFR mutations, for example, proteogenomic analysis can reveal key downstream effects, such as phosphorylation in the PI3K/AKT pathway, informing more tailored therapeutic decisions [20]. KRAS mutations often lead to downstream protein-level changes, such as the overactivation of signaling pathways like MAPK/ERK or PI3K/AKT. These changes can create specific vulnerabilities that make the tumor susceptible to targeted inhibitors [21]. By combining proteomic insights (which show how proteins and pathways are affected) with genomic data (which reveals specific mutations), clinicians can develop treatment plans that address both the genetic mutations and their resulting biochemical consequences, leading to a more comprehensive and effective approach to care.

Proteogenomic analyses have also uncovered metabolic vulnerabilities and dysregulated signaling pathways in thoracic cancers, which are crucial for precision treatment strategies. For instance, the GLUT1 glucose transporter, often overexpressed in aggressive lung tumors, indicates a dependence on glucose metabolism for growth. Consequently, metabolic inhibitors like GLUT1 inhibitors are being investigated alongside surgery to improve outcomes [22].

The integration of multi-omic data may also support surgical planning by providing detailed insights into tumor behavior, potentially guiding decisions between segmentectomy and lobectomy. In the future, findings from proteomic analysis showing low levels of metastasis-associated biomarkers, such as E-cadherin or matrix metalloproteinases (MMPs) [23,24], might validate the choice of a segmentectomy, if technically feasible, to preserve lung function. Conversely, if proteogenomic analysis reveals aggressive biomarkers, such as high MMP activity, VEGF overexpression, or TP53 mutations, a lobectomy might be recommended to ensure radicality of resection. However, the use of these multi-omic approaches in surgical decision-making is still evolving and requires further validation.

These findings underscore the value of combining multi-omic data to form a cohesive and comprehensive strategy for managing thoracic cancers [Table 2]. By understanding both metabolic and signaling pathways at a molecular level, treatment can be personalized, resulting in more effective responses, lower recurrence rates, and improved overall patient outcomes.

## 4. Understanding Tumor Heterogeneity Through Multi-Omic Profiling

One of the key challenges in managing lung cancers within the broader field of thoracic oncology is the significant degree of tumor heterogeneity [25]. Such heterogeneity complicates treatment, as even a single tumor can contain multiple distinct cellular subpopulations with varied genetic and phenotypic characteristics. To better address this complexity, multi-omic profiling—integrating different layers of molecular information—is increasingly employed. Through genomics, we understand the genetic basis; transcriptomics reveals gene expression patterns; and proteomics identifies functional protein markers. By mapping this intricate molecular landscape, clinicians gain deeper insights into tumor architecture.

Several studies have utilized single-cell RNA sequencing to identify subpopulations within NSCLC that harbor resistance-associated transcripts and immune-evading phenotypes [26,27]. Proteomics-based investigations have correlated protein-level changes, such as kinase overexpression, with response or resistance to targeted therapies [28]. Concerning epigenomics, alterations in DNA methylation patterns, histone modifications, and chromatin remodeling can profoundly affect gene expression, influencing key processes such as cell proliferation, metastasis, and immune evasion [29]. For instance, hypermethylation of tumor suppressor genes and changes in histone acetylation can promote a more invasive tumor phenotype and correlate with advanced disease stages and poorer clinical outcomes. Integrating transcriptomic and epigenomic findings may allow distinguishing indolent tumors from those with aggressive methylation patterns that predict early relapse, potentially guiding decisions to more extensive resections.

While emerging evidence suggests that heterogeneous subclones identified through multi-omic profiling can be therapeutically targeted, potentially increasing the precision of both systemic therapy and surgical intervention, these approaches remain largely in their infancy. Although this holistic methodology holds promise for devising effective adjuvant strategies, ensuring thorough disease eradication, and fostering better recovery, significant challenges persist. As multi-omic technologies mature, become more accessible, and gain standardization, their role in thoracic surgical oncology is poised to expand. In the meantime, clinicians and researchers must continue to refine these tools, validate their clinical utility through robust studies, and carefully integrate them into practice to truly realize the envisioned improvements in patient stratification and durable treatment outcomes.

## 5. Monitoring Disease Progression and Detecting Minimal Residual Disease

Monitoring disease progression and detecting MRD are critical steps in improving long-term outcomes. Circulating tumor DNA (ctDNA) profiling post-surgery can indicate residual disease or micrometastasis before radiological recurrence, enabling earlier and more effective intervention [30]. Detecting emerging resistance mutations through ctDNA analysis enables timely therapeutic adjustments, potentially maintaining treatment efficacy and preventing disease progression [31]. Such early detection is not merely theoretical—prospective studies have demonstrated that ctDNA can identify emerging resistance mutations, such as secondary EGFR alterations, well in advance of imaging-based progression [32].

In practice, these findings mean that if ctDNA levels rise and resistance mutations appear, clinicians can modify therapy promptly—for instance, switching from a first-generation EGFR inhibitor to a third-generation agent—potentially delaying clinical relapse. Some early-phase clinical trials are even testing the feasibility of using ctDNA dynamics to stratify adjuvant therapies, ensuring intensified treatment for molecularly active disease and sparing low-risk patients from unnecessary interventions. For example, the ongoing MERMAID-2 trial is designed to enroll patients who have undergone complete resection of Stage II–III NSCLC and received standard-of-care chemotherapy. The presence of MRD is assessed by ctDNA testing, and patients who test positive are then randomized to receive durvalumab or placebo. As these trials report out, guidelines may evolve to incorporate ctDNA-driven MRD assessment, providing thoracic surgeons and medical oncologists with a powerful tool to fine-tune postoperative management in real time.

To further enhance the detection of minimal residual disease, the use of liquid biopsies has proven pivotal. Liquid biopsies generally refer to blood-based tests that detect tumor-derived biomarkers such as ctDNA and circulating tumor cells (CTCs). Unlike traditional tissue biopsies, which are invasive and limited to a single time point, liquid biopsies are minimally invasive and can be performed repeatedly, providing a dynamic, real-time view of the tumor’s molecular evolution [33]. By integrating data from ctDNA and circulating tumor cells (CTCs), clinicians gain valuable insights into the changing molecular landscape. As resistance mutations arise—such as secondary EGFR mutations or alterations in downstream signaling pathways—ctDNA analysis allows clinicians to preemptively modify therapy, selecting next-generation inhibitors or combination regimens before overt clinical or radiographic progression occurs. This proactive approach can help preserve response rates, delay recurrence, and potentially extend overall survival. However, there is a pressing need to advance the development of biomarkers with high sensitivity and specificity for the early detection of lung cancer. Additionally, a comprehensive prognostic biomarker has yet to be identified. The adoption of circulating biomarkers in routine clinical practice faces significant challenges, including the absence of standardized detection methods, limited accessibility, high costs, and uncertainty regarding cutoff levels. More robust evidence is required to transition liquid biopsy from the research phase to clinical application. Despite these obstacles, its potential role in the diagnosis and treatment of lung cancer remains undeniable.

As these technologies advance, incorporating MRD monitoring into standard postoperative management may become the norm. Combined with molecular imaging and other diagnostic tools, ctDNA-driven assessments could help clinicians deliver truly personalized, adaptive treatment regimens that evolve in tandem with the tumor’s molecular profile.

## 6. Molecular Imaging in Thoracic Surgical Oncology

Molecular imaging plays a crucial role in the era of precision medicine by providing detailed insights into tumor biology that go beyond the capabilities of conventional imaging. Techniques such as positron emission tomography (PET), single-photon emission computed tomography (SPECT), and novel radiotracers are transforming the detection, staging, and treatment planning of thoracic cancers.

PET imaging, particularly when combined with computed tomography (PET/CT), enables the visualization of metabolic activity within tumors, which can be used to assess tumor aggressiveness and monitor response to therapy. For instance, fluorodeoxyglucose (FDG)-PET is frequently used to identify metabolically active tumor sites, providing critical information for determining the extent of disease and guiding surgical resection strategies. PET imaging also facilitates the evaluation of treatment response in real time, allowing clinicians to adapt therapeutic plans based on the metabolic changes observed [34].

In addition, molecular imaging can employ targeted radiotracers that bind to specific tumor markers. For example, molecular imaging also plays a pivotal role in the management of thoracic neuroendocrine tumors, such as carcinoids. These tumors commonly overexpress somatostatin receptors, enabling the use of radiolabeled somatostatin analogs for diagnostic imaging and therapeutic planning. For instance, gallium-68 (Ga-68)-labeled somatostatin receptor PET/CT imaging agents, such as Ga-68 DOTATOC, have become invaluable tools in evaluating carcinoid tumors. Studies have shown that somatostatin receptor imaging not only enhances the detection of primary lesions and metastases that might be missed by conventional CT or MRI but can also influence surgical decision-making by more accurately delineating the extent of disease [35]. Identifying previously unrecognized metastatic localizations, for example, may alter a planned curative resection to a more conservative approach or prompt consideration of systemic therapies. Conversely, confirming a localized, receptor-avid lesion could support a more aggressive surgical strategy.

Beyond diagnosis and staging, somatostatin receptor imaging paves the way for targeted radionuclide therapies, such as lutetium-177 (Lu-177) DOTATATE peptide receptor radionuclide therapy (PRRT). This treatment exploits the same somatostatin receptor overexpression, delivering cytotoxic radiation directly to tumor cells while sparing normal tissues. Patients with carcinoid tumors that demonstrate high receptor avidity on imaging are often prime candidates for PRRT, which has shown efficacy in controlling disease progression and alleviating symptoms [36].

Beyond established modalities like FDG-PET and somatostatin receptor imaging, a new generation of molecular imaging tracers is emerging to further refine thoracic surgical oncology. Notably, epidermal growth factor receptor (EGFR)-targeted imaging agents are under investigation, aiming to visualize EGFR-driven tumors and their response to targeted therapies [37]. Early-phase studies using radiolabeled EGFR inhibitors (e.g., 11C-erlotinib or other EGFR-binding radiotracers) have demonstrated the feasibility of mapping receptor expression in vivo [38]. These agents have the potential to differentiate active, EGFR-dependent tumor regions from areas rendered fibrotic by neoadjuvant treatments. For thoracic surgeons, such information could help assess resectability and delineate resection margins more precisely, ensuring that residual, metabolically active tumor tissue is removed while preserving healthy lung parenchyma. As research advances and these tracers become more readily available, they may join the armamentarium of molecular imaging tools that guide personalized surgical planning, especially in cases where EGFR mutations play a key role in tumor biology. Over time, the integration of EGFR-targeted imaging into multidisciplinary treatment protocols may inform both immediate operative decisions and longer-term strategies for systemic therapy and surveillance.

As molecular imaging and multi-omic insights refine our understanding of tumor boundaries, infiltration patterns, and molecular vulnerabilities, surgical planning can become more nuanced. In cases where imaging reveals that the tumor involves critical vascular structures, such as the pulmonary artery, detailed preoperative assessment may guide the use of reconstructive techniques to preserve lung parenchyma. Rather than defaulting to a pneumonectomy, surgeons may opt for pulmonary artery reconstruction or bronchovascular sleeve resections, effectively removing the tumor while maintaining adequate pulmonary function [39,40]. Such approaches can facilitate parenchyma-sparing surgery and avoid the morbidity associated with pneumonectomy. Integrating these surgical advances into the broader precision oncology framework exemplifies how molecular diagnostics, imaging, and technical expertise converge to enhance both oncologic outcomes and quality of life for patients with complex thoracic malignancies.

These advances enable a more tailored approach to both surgical and systemic therapy by providing high-resolution, functional data that complement anatomical imaging.

Molecular imaging also contributes significantly to radiotherapy planning [41]. By defining tumor boundaries more precisely and identifying areas with high proliferative activity, radiation oncologists can deliver targeted doses that maximize tumor control while minimizing exposure to healthy tissues. This precision is vital in thoracic oncology, where the proximity of tumors to critical structures like the heart and major vessels necessitates careful treatment planning.

Although current guidelines recommend imaging-based staging and response assessment, the use of advanced molecular imaging tracers is still not universally standardized and may vary depending on regional availability and expertise. Further research and consensus-building are needed to incorporate these novel imaging modalities into routine clinical decision-making.

In the near future, novel radiotracers targeting other oncogenic pathways or metabolic vulnerabilities may enter clinical trials, further aligning imaging findings with omics-driven insights. Such advances could refine patient selection for innovative surgical approaches, ensuring that evolving imaging tools continuously enhance and complement the precision-driven landscape of thoracic surgical oncology.

## 7. Challenges, Future Directions, and Ethical Considerations

While integrating oncogenomic and multi-omic data into thoracic surgical oncology offers new possibilities, significant challenges persist. Standardizing methodologies, proving cost-effectiveness, and refining guidelines remain critical steps before these technologies become routine. NCCN and ESMO currently recommend molecular testing for key driver mutations and immunotherapy markers in NSCLC, yet they do not comprehensively incorporate multi-omic profiling strategies, reflecting the nascent state of these approaches.

High costs and technical requirements also limit access, underscoring the need for advancements that make precision medicine more accessible. Education and training for clinicians in interpreting complex molecular data are also vital in fully leveraging these tools in practice. Future research should focus on developing predictive algorithms that synthesize multi-omic data to enhance decision-making accuracy. Fostering interdisciplinary collaboration between oncologists, surgeons, and bioinformaticians is essential to close the gap between data generation and its clinical application. By refining these integration techniques, precision medicine can be made more practical, ultimately improving patient outcomes on a larger scale.

In addition to multi-omic complexities, the use of circulating tumor DNA (ctDNA) as a biomarker in thoracic cancers poses its own set of challenges. While ctDNA monitoring can detect minimal residual disease (MRD) earlier than imaging, questions persist regarding assay sensitivity, specificity, and standardization. Different ctDNA platforms and analytical methods can yield variable results, complicating comparisons across studies and centers. Establishing uniform cutoffs for ctDNA positivity and identifying which mutations best predict recurrence risk remain active areas of research. Moreover, it is unclear how frequently ctDNA testing should be performed, how results should guide therapy modifications, and whether all patient subsets benefit equally from MRD-driven interventions. Further research is required to determine when and how ctDNA monitoring should be implemented, ensuring that it informs more precise surgical planning, tailors systemic therapies effectively, and optimizes long-term follow-up protocols.

The application of precision medicine extends beyond scientific and clinical challenges to encompass a range of ethical considerations. Detailed genomic and proteomic analyses produce vast amounts of patient-specific data, raising questions about data privacy, informed consent, and the potential for genetic discrimination. Patients must understand not only the immediate therapeutic implications of their molecular results but also the long-term ramifications of having their genetic and molecular information stored and potentially shared among healthcare providers and researchers. Transparent communication about data handling, security measures, and the de-identification of sensitive information is essential to maintain patient trust.

Another ethical concern is equitable access to precision medicine technologies. Advanced molecular tests, specialized imaging tracers, and targeted therapies often concentrate in well-resourced academic centers, creating disparities in care. Patients in underserved or rural areas may have limited opportunities to benefit from cutting-edge diagnostics and treatments, widening health inequality. Addressing this issue may involve policy initiatives, public–private partnerships, and telemedicine solutions that extend molecular testing and expert interpretation to broader populations.

Furthermore, the integration of proteomic and genomic data into clinical decision-making may uncover incidental findings—genetic variants or molecular markers unrelated to the patient’s current thoracic malignancy but potentially significant for other health risks. Deciding whether and how to disclose these findings involves careful ethical reasoning. Balancing patient autonomy, the duty to prevent harm, and the risk of creating undue anxiety is complex, and no universal standard has emerged. Professional societies and ethics committees will likely need to provide guidance on best practices, ensuring clinicians act responsibly and respectfully when dealing with these scenarios.

Continued efforts in research, education, and collaboration will be critical to overcoming current obstacles, ensuring that precision medicine reaches its full potential in improving cancer care and delivering more personalized treatment strategies.

## 8. Conclusions

Integrating oncogenomic, proteomic, transcriptomic, epigenomic, and other multi-omic data streams is rapidly transforming the landscape of thoracic surgical oncology, influencing every step of the diagnostic and therapeutic pathway (Figure 1). By analyzing the molecular underpinnings of each patient’s tumor—encompassing mutation profiles, pathway activations, immune signatures, and epigenetic alterations—clinicians can now tailor both systemic and surgical approaches with a level of precision previously unattainable.

These molecular analyses seamlessly integrate with advanced imaging modalities—such as FDG-PET, EGFR-targeted tracers, and somatostatin receptor imaging for neuroendocrine tumors—leading to more accurate delineation of tumor borders, better identification of candidates for parenchyma-sparing procedures, and opportunities for bronchovascular or vascular reconstruction techniques to avoid extensive resections like pneumonectomy. Circulating biomarkers, including ctDNA, add yet another dimension by detecting minimal residual disease and emerging resistance mutations early, enabling proactive adjustments in postoperative therapies that may improve long-term disease control.

Despite these promising developments, significant hurdles remain. Technical challenges in assay standardization, uncertainties regarding the clinical utility of new biomarkers, and cost-effectiveness debates must be resolved before these approaches see universal adoption. Ethical considerations—such as safeguarding patient privacy, ensuring equitable access to advanced diagnostics, and responsibly interpreting incidental findings—further complicate clinical implementation.

Looking ahead, multidisciplinary collaboration, robust clinical trials, and strong real-world evidence will be crucial in translating ongoing research into guideline-endorsed best practices. Through continued refinement of these approaches, validation of their clinical benefit, and the establishment of ethical, patient-centered frameworks, thoracic surgical oncology can move toward a future where every patient receives personalized, evidence-based, and minimally morbid care. In this evolving era, precision medicine stands not merely as a scientific aspiration, but as a practical, attainable goal—one that steadily improves outcomes and shapes the next generation of thoracic oncology interventions.

## Figures and Tables

**Figure 1 cancers-17-00115-f001:**
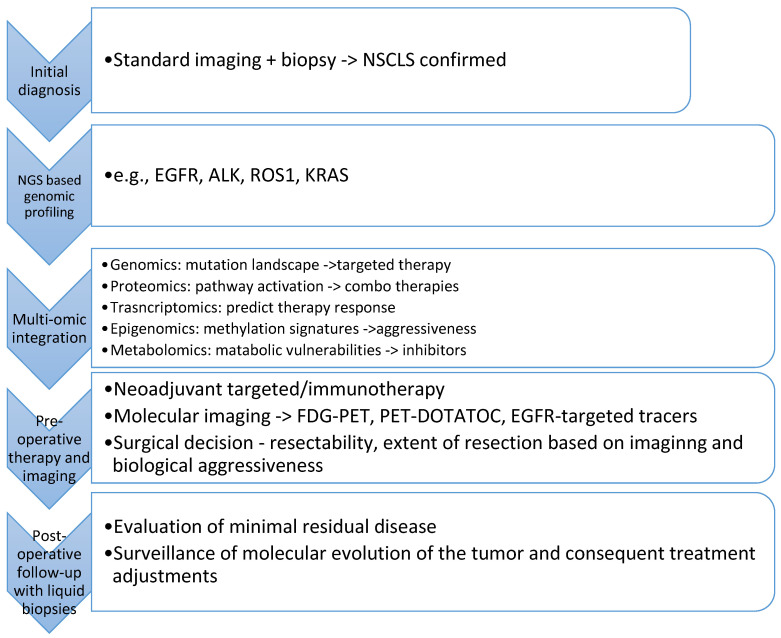
Integrating precision medicine into the thoracic surgical oncology care pathway.

**Table 1 cancers-17-00115-t001:** Recent clinical trials that showed the benefit of immunotherapy with nivolumab in the neoadjuvant setting.

Study	Design and Patient Population	Interventions	Key Outcomes
**CheckMate 816**	Phase III, patients with resectable stage IB (≥4 cm)—IIIA NSCLC without EGFR/ALK alterations.	Neoadjuvant nivolumab + platinum-based doublet chemotherapy vs. chemotherapy alone.	**EFS:** 31.6 months with nivolumab vs. 20.8 months with chemotherapy alone; **pCR:** 24% with nivolumab vs. 2.2% with chemotherapy alone.
**NADIM**	Phase II single arm, patients with resectable stage IIIA NSCLC.	Neoadjuvant nivolumab + chemotherapy followed by adjuvant nivolumab.	**24-month PFS:** 77.1% **pCR**: 63%.
**NADIM II**	Phase II, patients with resectable stage IIIA or IIIB NSCLC.	Neoadjuvant nivolumab + chemotherapy, followed by adjuvant nivolumab vs. platinum-based chemotherapy alone.	**pCR:** 37% with nivolumab + chemotherapy vs. 7% with chemotherapy alone; **24-month OS:** 85% with nivolumab + chemotherapy vs. 63.6% with chemotherapy alone.

EFS: event-free survival; PFS: progression-free survival; pCR: pathological complete response; OS: overall survival.

**Table 2 cancers-17-00115-t002:** Overview of multi-omic approaches in thoracic oncology.

Omics Technology	Role in Thoracic Oncology	Clinical Applications	Examples
**Genomics**	Identifies driver mutations, gene fusions, and actionable targets.	Guides targeted therapies (EGFR, ALK inhibitors), influences surgical resectability decisions.	EGFR/ALK/ROS1 mutation testing.
**Proteomics**	Detects protein-level changes, post-translational modifications, and pathway activations.	Tailors combination therapies, refines surgical approach based on pathway vulnerabilities.	PI3K/AKT pathway activation in EGFR-mutated tumors.
**Transcriptomics**	Analyzes gene expression patterns, identifies active pathways and immune-related signatures.	Enables comprehensive tumor profiling for precision treatment, predicts response to immunotherapy.	Single-cell RNA-seq identifying immune-evading subclones and guiding immunotherapy addition.
**Epigenomics**	Examines DNA methylation, histone modifications influencing gene expression.	Stratifies tumors by aggressiveness, potential for metastasis, and long-term recurrence risk, refining surgical planning.	Hypermethylation of tumor suppressor genes associated with poor outcomes and need for more aggressive surgical intervention.
**Metabolomics**	Reveals metabolic dependencies and vulnerabilities.	Integrates metabolic inhibitors with surgery to improve outcomes.	GLUT1 inhibitor studies showing synergy with surgical resection.

## Data Availability

Not applicable.

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
