# Peer review of "Advancing Thoracic Surgical Oncology in the Era of Precision Medicine"

_cancers, 2025, doi:10.3390/cancers17010115_

Round 1
Reviewer 1 Report
Comments and Suggestions for Authors
Dear Authors,
The manuscript content is good for good readers, but I suggest some strong revisions for your scientific readers.

Author Response
Dear Reviewer,
Thank you for your thoughtful and detailed feedback on our manuscript. We greatly appreciate the time you invested in providing such constructive suggestions, which have significantly guided us in refining the clarity, specificity, and coherence of our work.
In response to your comments on the section addressing tumor heterogeneity and multi-omic profiling, we have carefully revised the text to more explicitly distinguish our focus on lung cancers within the broader field of thoracic oncology. We have also expanded the explanation of each omic layer—genomics, transcriptomics, and proteomics—and clarified how identifying resistant subpopulations informs adaptive treatment strategies. Additionally, the significance of single-cell RNA sequencing has been emphasized, illustrating its crucial role in pinpointing resistant tumor cell populations and tailoring individualized therapies. By incorporating concrete examples—such as the identification of overexpressed kinases through proteomics—we now provide more tangible benefits of multi-omic insights. We have also highlighted the direct link between targeting these resistant clones and the potential for reducing recurrence rates and improving long-term survival, underscoring the integrative and holistic nature of the approach.
Similarly, in the section on monitoring disease progression and detecting minimal residual disease, we have restructured and clarified certain sentences to enhance readability and precision. We have introduced the term “comprehensive genomic and molecular analysis” to help readers unfamiliar with multi-omic concepts, and we have removed redundant terminology for conciseness. Furthermore, we have tempered statements regarding the ultimate clinical benefit of liquid biopsies and provided a smoother transition into the discussion of their utility in MRD detection. Additional context has been added to explain how early identification of resistance mutations in ctDNA enables timely adjustments to therapy.
Overall, we have expanded key concepts, refined technical wording, and integrated your recommendations to improve both the accessibility and scientific rigor of the manuscript. We believe these revisions address your concerns and result in a more informative, coherent, and reader-friendly text. Your guidance has proven invaluable in shaping a more robust and insightful article, and we thank you once again for your constructive review.
Sincerely
Reviewer 2 Report
Comments and Suggestions for Authors It is accept, but the content is too short, so it is suitable as a short communication type paper.Author Response
Dear Reviewer,
Thank you for your feedback and for your positive assessment of our work. We appreciate your suggestion that the manuscript, as initially presented, might be more suitable as a short communication due to its brevity.
In light of your comment, we have substantially expanded and enriched the manuscript. We have added greater depth to each section, provided more detailed discussions of multi-omic approaches, and included new references and practical examples to ensure the content offers a more comprehensive and instructive perspective. This expanded version now approaches the length and level of detail expected for a full-length review article, rather than a short communication.
We trust these enhancements address your concerns and provide readers with a more extensive and in-depth resource on how precision medicine is transforming thoracic surgical oncology. Thank you again for your insights, which have contributed to making the manuscript more complete and informative.
Sincerely
Reviewer 3 Report
Comments and Suggestions for Authors
The presented article offers a comprehensive and well-structured overview of the current developments in the treatment of lung tumors within the context of modern precision medicine. The authors highlight already established new technologies such as Next-Generation Sequencing (NGS), immunotherapies and molecular imaging for the development of personalized treatment strategies. Furthermore, the article presents the perspectives of multi-omic approaches, which not only create genomic but also proteomic and metabolic profiles of tumors. The techniques discussed are supported by current literature. Recent clinical studies on the use of immunotherapies in neoadjuvant treatment are presented in a tabular format.
The article is aimed at both medical researchers and clinicians dealing with the latest developments in lung cancer treatment. Overall, the article is well-structured and informative, but at times it remains somewhat vague, preventing it from serving as a "vade mecum" for clinical colleagues.
Therefore, I would like to ask the authors to consider the following suggestions for additions, which I believe would enhance the article:
- The article could benefit from the inclusion of concrete case scenarios or imaging data from clinical practice.
- Precision medicine, and particularly the use of genomic and proteomic data in treatment, raises ethical questions that could be addressed in more detail in the article.
- References to the current guidelines would show which of these technologies are already applied as standard tools and which questions remain unresolved. By discussing the latest technological advancements in the context of these guidelines, the article could highlight where further research is needed to integrate these approaches into standard treatment plans.
Author Response
Dear Reviewer,
Thank you for your careful reading of our manuscript and for your encouraging remarks about its structure and comprehensiveness. We appreciate your observation that, at times, the content remained somewhat vague, and we are grateful for your specific suggestions on how to enrich the text.
In response to your comments, we have taken several steps to provide a more practical and clinically relevant perspective. First, we have included more concrete examples that link molecular imaging findings and multi-omic data to real-world surgical decisions. While we have not included direct patient imaging data, we have contextualized how these imaging modalities and molecular insights are applied in actual clinical settings, thereby making the discussion more tangible and clinically useful.
Additionally, we have expanded our examination of the ethical dimensions of precision medicine. We now provide a more in-depth discussion of issues such as data privacy, the potential for genetic discrimination, and the equitable distribution of advanced testing and targeted therapies. By integrating these considerations, we hope to provide readers with a fuller understanding of the practical and moral complexities that arise as genomics, proteomics, and other molecular analyses become integral to thoracic surgical oncology.
We have also incorporated references to current clinical guidelines and standard-of-care recommendations, highlighting which technologies are already endorsed by major organizations (e.g., NCCN, ESMO) and where emerging omics-driven approaches still await validation. This added context should help clarify which aspects of precision medicine are currently implemented and which remain in development, emphasizing areas that require further research and consensus-building before they become widely adopted in everyday practice.
By addressing these points, we believe we have moved closer to creating a resource that clinicians and researchers can rely upon not only for an overview of the latest technologies but also as a practical guide. We trust these revisions strengthen the manuscript’s utility, ensuring it better serves both the scientific community and clinical colleagues seeking to integrate precision medicine into thoracic surgical oncology.
Thank you once again for your valuable suggestions and for helping us improve the clarity and applicability of our work.
Sincerely
Reviewer 4 Report
Comments and Suggestions for Authors
The manuscript, “The Role of Thoracic Surgical Oncology in the Era of Precision Medicine” underscores how advancements in precision medicine are transforming thoracic surgical oncology. The review has great points and addresses the how emerging technique may help the conventional treatment for lung cancer. However, the review needs to be further detailed and refined as follows:
I suggest the author to make change in the title of the paper. For example, “Advancing Thoracic Surgical Oncology in the Era of Precision Medicine” or “Revolutionizing Thoracic Surgical Oncology: in the Era of Precision Medicine" or "Precision Medicine and Its Impact on Thoracic Surgical Oncology: A New Frontier in Cancer Treatment" or author may come up with a new one.
Table 2: In the context of multiomic studies, the authors primarily discuss genomic and proteomic data. However, it may be worthwhile to explore whether other omics layers, such as epigenomics or transcriptomics, have been explored in thoracic oncology.
Under “4. Understanding Tumor Heterogeneity through Multi-Omic Profiling”, authors mentioned how single-cell RNA sequencing and proteomics, can help “reveal unique protein markers that can be therapeutically targeted”. Adding few studies that have explored these areas would provide stronger evidence for the discussion.
Overall, the review is well-written; however, it reads more like a brief overview rather than an in-depth analysis. To enhance the depth of the review, I suggest that the authors provide more detailed information in each section. For instance, the review mentions frequently “multi-omic profiling” or next-generation sequencing. It would be valuable if the authors could incorporate a broader range of omics studies conducted on thoracic cancers and explore how these studies contribute to precision medicine and impact surgical interventions.
Under the section "Challenges and Future Directions," the authors primarily focus on the challenges associated with multi-omic profiling. However, it would be beneficial to also address the challenges related to other aspects mentioned earlier, such as the use of ctDNA as biomarkers in thoracic cancers. This area remains an ongoing field of research and warrants further discussion in the context of the review.
It would be great of authors can make a figure or two to that briefly summaries all the points mentioned to further enrich the review.
Add keyword “next-generation sequencing” or “Multi-Omic profiling”
Author Response
Dear Reviewer,
Thank you for your insightful comments and recommendations on how to strengthen our manuscript. We appreciate your acknowledgment of the article’s potential and have taken your suggestions into careful consideration.
First, we have adopted your advice to revise the title, choosing one that better encapsulates the evolving landscape of thoracic surgical oncology in the era of precision medicine. We have also integrated the keyword “next-generation sequencing” and “multi-omic profiling” to ensure the manuscript is more discoverable and aligned with the terminology frequently used in this field.
In Table 2, we have expanded our discussion beyond genomics and proteomics to include transcriptomics and epigenomics, describing how these additional omics layers are increasingly explored in thoracic oncology research. This provides a more holistic view of how multiple molecular dimensions contribute to refining patient stratification, guiding neoadjuvant or adjuvant therapy selection, and informing surgical decisions.
Building on this, we have incorporated references to specific studies in the section on “Understanding Tumor Heterogeneity through Multi-Omic Profiling” to illustrate how single-cell RNA sequencing and proteomics have already begun to identify tangible therapeutic targets. By adding concrete literature-based examples, we believe this section now offers stronger evidence and greater depth.
Your suggestion to provide a more in-depth analysis rather than a brief overview has prompted us to expand various sections, adding more detailed discussions of transcriptomic, epigenomic, and metabolomic findings and explaining how these data translate into clinically meaningful interventions. We have also addressed challenges beyond multi-omic profiling—specifically the complexities surrounding ctDNA as a biomarker—and highlighted the need for assay standardization, sensitivity thresholds, and integration into treatment algorithms.
Finally, we have proposed a figure summarizing the integration of diverse omics layers, imaging findings, and ctDNA monitoring into the clinical decision-making pathway. Such a figure aims to offer readers a visual roadmap, making the review more practical and user-friendly.
We believe these enhancements create a more comprehensive, in-depth, and clinically applicable review. Thank you again for your constructive critique and for guiding us toward a more valuable resource for both researchers and clinicians.
Sincerely